# Arginine Dysregulation and Myocardial Dysfunction in a Mouse Model and Children with Chronic Kidney Disease

**DOI:** 10.3390/nu15092162

**Published:** 2023-04-30

**Authors:** Loretta Z. Reyes, Pamela D. Winterberg, Roshan Punnoose George, Michael Kelleman, Frank Harris, Hanjoong Jo, Lou Ann S. Brown, Claudia R. Morris

**Affiliations:** 1Division of Pediatric Nephrology, Emory University School of Medicine, Atlanta, GA 30322, USA; 2Children’s Healthcare of Atlanta, Atlanta, GA 30322, USA; 3Department of Pediatrics, Emory University School of Medicine, Atlanta, GA 30322, USA; 4Division of Cardiology, Emory University School of Medicine, Atlanta, GA 30322, USA; 5Division of Pediatric Emergency Medicine, Emory University School of Medicine, Atlanta, GA 30322, USA

**Keywords:** arginine bioavailability, arginase, nitric oxide, cardiovascular disease, chronic kidney disease

## Abstract

Cardiovascular disease is the leading cause of death in chronic kidney disease (CKD). Arginine, the endogenous precursor for nitric oxide synthesis, is produced in the kidneys. Arginine bioavailability contributes to endothelial and myocardial dysfunction in CKD. Plasma from 129X1/SvJ mice with and without CKD (5/6th nephrectomy), and banked plasma from children with and without CKD were analyzed for amino acids involved in arginine metabolism, ADMA, and arginase activity. Echocardiographic measures of myocardial function were compared with plasma analytes. In a separate experiment, a non-specific arginase inhibitor was administered to mice with and without CKD. Plasma citrulline and glutamine concentrations correlated with multiple measures of myocardial dysfunction. Plasma arginase activity was significantly increased in CKD mice at 16 weeks vs. 8 weeks (*p* = 0.002) and ventricular strain improved after arginase inhibition in mice with CKD (*p* = 0.03). In children on dialysis, arginase activity was significantly increased vs. healthy controls (*p* = 0.04). Increasing ADMA correlated with increasing RWT in children with CKD (r = 0.54; *p* = 0.003). In a mouse model, and children, with CKD, arginine dysregulation correlates with myocardial dysfunction.

## 1. Introduction

Cardiovascular disease (CVD) is the leading cause of death in patients with chronic kidney disease (CKD). Children and young adults with childhood-onset CKD have a nearly 30-fold increased risk of mortality compared to their peers, and CVD accounts for >25% of deaths [1,2,3]. The etiologies for CKD differ in children compared to adults who typically develop CKD due to diabetes mellitus or hypertension (HTN). While some children with CKD develop HTN (a known risk factor for CVD), children with CKD lack many of the other traditional cardiovascular risk factors seen in adults, therefore the mechanisms underlying this increased cardiovascular mortality risk are incompletely understood. Uremic cardiomyopathy, characterized by left ventricular hypertrophy (LVH), diastolic dysfunction and impaired ventricular strain, describes some of the pathologic myocardial changes that occur in CKD. LVH, the most common manifestation of uremic cardiomyopathy, is highly prevalent in children with CKD [4,5,6] and predicts mortality in adults with CKD [7,8].

Endothelial dysfunction, due to reduced production or reduced bioavailability of nitric oxide (NO), is a known contributor to the development of CVD [9,10,11]. CKD is considered to be a NO deficiency state [12,13,14,15] and endothelial dysfunction occurs throughout the stages of CKD [16,17,18]. The relationship between arginine bioavailability, NO biosynthesis and cardiovascular disease is complex. Arginine is a nutritional amino acid and the obligate substrate for NO biosynthesis (Figure 1) and its availability in CKD for NO synthesis may be impaired by reduced endogenous arginine synthesis, diversion of arginine to other metabolic pathways and impaired intracellular delivery of arginine to NOS enzymes [19,20,21]. The arginase enzyme is a critical regulator of NO synthesis by competition with the NO synthase enzyme for the shared substrate, arginine. Increased arginase activity has been shown to contribute to endothelial dysfunction in multiple disease states including hypertension, atherosclerosis, heart failure, sickle cell disease and diabetes [22,23,24,25] and arginase inhibition has shown beneficial effects on endothelial and cardiovascular function [26,27]. Asymmetric dimethylarginine (ADMA), the methylated arginine derivative, is a competitive inhibitor of NOS and an established biomarker of cardiovascular risk. ADMA is known to accumulate as kidney failure progresses [14,28] and contributes to reduced NO bioavailability.

Amino acid metabolism, including that of arginine and its derivatives, is known to be abnormal in CKD [29,30,31], however, studies investigating associations between plasma arginine concentration and cardiovascular complications have had mixed results. Interestingly, global arginine bioavailability ratio (GABR), which is defined as the ratio of plasma arginine/(ornithine + citrulline), has been found to be a more reliable biomarker of arginine bioavailability than arginine concentration alone [25] and is predictive of increased cardiovascular risk [25,32]. Low GABR has been shown to represent an independent risk factor for cardiovascular morbidity and mortality in different patient populations e.g., low GABR correlated with pulmonary HTN severity in patients with sickle cell disease [25,33,34] and thalassemia [35], mortality in patients with sickle cell disease [25,36], and major adverse cardiovascular events, including mortality in adults with coronary artery disease [32,37,38].

In this study, our overarching hypothesis is that dysregulated arginine metabolism is associated with myocardial dysfunction in CKD. To investigate this, we correlated global arginine bioavailability, amino acids involved in arginine metabolism and arginase activity with measures of myocardial structure and function in the partial nephrectomy mouse model of CKD and in children with CKD. We also aimed to determine the effects of arginase inhibition on myocardial dysfunction in a mouse model of CKD.

## 2. Materials and Methods

### 2.1. Murine Experiments:

In 2 separate experiments, mice with and without CKD underwent non-invasive blood pressure measurements and echocardiograms prior to their endpoints (Appendix A). The first cohort (CKD *n* = 9, control *n* = 9) was euthanized at 8 weeks CKD at which time blood was collected for plasma and heart tissue was harvested and stored at −80 °C for subsequent tissue analysis. The second cohort (CKD *n* = 10, control *n* = 7) was euthanized at 16 weeks; blood and tissue were collected as per the 8-week cohort. In a separate series of experiments, mice with (*n* = 5) and without (*n* = 5) CKD received treatment with the oral, reversible, potent arginase inhibitor CB-280 sourced from Calithera Biosciences (https://www.calithera.com/; accessed on 18 November 2020). At 6 weeks of CKD, echocardiograms were performed, and then mice were gavage fed 300 mg/kg CB-280 twice daily for 5 days. On completion of treatment, echocardiograms were repeated, and mice were euthanized.

#### 2.1.1. Mouse Model of CKD

Prior reports have demonstrated that the partial nephrectomy mouse model of CKD recapitulates many of the cardiovascular comorbidities of CKD including systemic hypertension, LVH, diastolic dysfunction and impaired ventricular strain [39]. Female mice have been shown to be less affected in CKD models; consequently, male mice were utilized in this study [40,41]. Male 129X1/SvJ mice (The Jackson Laboratory, Bar Harbor, ME, USA) age 5–6 weeks were randomly assigned to undergo five-sixth nephrectomy (5/6 Nx) surgery (first cohort *n* = 9; second cohort *n* = 10) or sham surgeries (first cohort *n* = 9; second cohort *n* = 7) under inhaled isoflurane (2%) anesthesia, in a two-staged approach as previously reported [39]. Briefly, in the first stage, the left kidney was exposed via flank incision, de-capsulated and the upper and lower poles were then resected via selective cauterization with a [42] high-temperature fine tip cautery (Geiger Medical Technologies, Council Bluffs, IA, USA). After 1 week of recovery, the entire right kidney was removed via a right flank incision. The time of the second surgery was considered the onset of CKD. Sham surgeries (occurring on the same days as 5/6 Nx) involved flank incision and exposure of the kidneys but no tissue removal. Mice were fed the 2018 Teklad Global 18% Protein Rodent Diet (Envigo, Madison, WI, USA) ad libitum. Peripheral blood was collected via facial vein at the predetermined endpoints of 8 weeks (cohort 1) and 16 weeks (cohort 2) after CKD onset and processed for plasma, then stored at −80 °C. After euthanasia by carbon dioxide asphyxiation, heart tissue was stored at −80 °C for future analyses. Renal function was assessed via plasma urea nitrogen (Kidney Profile Plus, Vetscan Diagnostic system, Abaxis) and via plasma cystatin c levels (R&D systems) since creatinine can be unreliable in mice [42]. All animal experiments were conducted in accordance with the National Institutes of Health Guide for the Care and Use of Laboratory Animals using protocols approved by Emory University Institutional Animal Care and Use Committee.

#### 2.1.2. Murine Echocardiography

Transthoracic echocardiography (Vevo2100, VisualSonics, Toronto) was performed under 1–2% isoflurane anesthesia at 8 weeks (cohort 1) and 16 weeks (cohort 2) of CKD. Heart rate was monitored simultaneously by electrocardiography and maintained at 450–500 beats per minute for the duration of the recording. Standard 2D echocardiographic measures of LV dimension were performed in the M-mode short and long-axis views. Relative wall thickness (RWT) was calculated as (septal wall thickness + posterior wall thickness divided by LV diastolic diameter = IVS; d + LVPW; d)/LVID; d as a measure of LVH. LV diastolic function was assessed in the pulse wave Doppler mode by measurement of the ratio of the LV transmitral early peak flow velocity to LV transmitral late peak flow velocity (E/A ratio) of 4–5 averaged cardiac cycles from at least two scans per mouse. Ventricular strain analyses were conducted using 2D speckle tracking software (VevostrainTM Analysis). B-mode images of 300 frames at >200 frames/second were acquired from parasternal long-axis views and global longitudinal strain (GLS) was calculated from anterior and posterior apical, mid and basal segments. GLS is most representative of myocardial shortening at the endocardium and is a measure of early myocardial functional changes in mice with CKD. All echocardiographic measures were conducted by the Emory + Children’s Pediatric Animal Physiology core; the surgeon was not blinded to the surgical status of animals.

#### 2.1.3. Murine Non-Invasive Blood Pressure Monitoring

Systolic and diastolic blood pressure in non-anesthetized mice was measured non-invasively using the BP-2000 tail-cuff system. Mice were acclimated to the restrainer by taking blood pressure measurements on 3 consecutive days prior to the BP assessment.

### 2.2. Human Experiments

#### 2.2.1. Pediatric CKD Subjects

Banked plasma from children and young adults with CKD aged 1–21 years, previously recruited under an institutional review board-approved tissue acquisition protocol (Immune Monitoring Protocol, Institutional Review Board #6248; Clinicaltrials.gov Identifier: NCT01283295) was utilized [43]. Children receiving hemodialysis or peritoneal dialysis were classified as end-stage renal disease (ESRD). Patients with active oncologic disease, systemic infection or those receiving ongoing therapy for autoimmune disease or transplant rejection were excluded. A single collection of whole blood, peripheral blood mononuclear cells (PBMC), serum, and plasma aliquots from each patient were stored in the Emory Transplant Center biorepository. Subjects with both banked plasma and clinically acquired echocardiograms in their hospital charts (within 6 months of plasma collection) available for analysis were included in the current study (Appendix A). The study was approved by Emory University and Children’s Healthcare of Atlanta Institutional Review Board #14-157, 5 May 2016.

#### 2.2.2. Pediatric Healthy Control Subjects

Banked plasma collected under the above tissue acquisition protocol was also utilized for controls. Healthy children and young adults with no obvious underlying inflammatory condition who presented for evaluation or follow-up of benign conditions (e.g., circumcision, solitary kidney or short stature) and were having a blood draw for pre-operative or monitoring reasons underwent recruitment into that protocol. Similarly, a single collection of whole blood, PBMC, serum, and plasma aliquots from each patient was stored in the Emory Transplant Center biorepository. Healthy subjects with adequate plasma available for the metabolites of interest were included in the study.

#### 2.2.3. Pediatric Echocardiography

For the CKD and ESRD patients included in the study (Appendix A), their clinically acquired echocardiograms were reviewed; no new echocardiograms were performed for this study. Standard of care clinically acquired echocardiograms were performed under the NCT01283295 protocol with IE33 Phillips ultrasound machine (Andover, MD, USA) using standard American Society of Echocardiography (ASE) guidelines. LV mass was calculated using the Devereux equation [44] and then indexed to height [45] using the published equation of LV mass index (LVMI) = LV mass in grams/height 2.7. LVMI was dichotomized into z-score <2 or ≥2 based on reported sex and age-related normative values [46] with LVH classified as LVMI z-score ≥2.

### 2.3. Tissue and Plasma Analyses

#### 2.3.1. Measurement of NO Metabolites (NO_x_), Amino Acids, and ADMA in Plasma

Total nitrate/nitrite concentration in plasma (NO_x_; mmol/L) was measured by a colorimetric assay based on the Greiss reaction (Cayman Chemical, Ann Arbor, MI, USA) as previously described [47,48]. Plasma amino acids (arginine, ornithine, citrulline, proline, glutamic acid, glutamine, and lysine) and ADMA were all quantified via liquid chromatography-tandem mass spectroscopy (LC-MS/MS) in mmol/L. All analyses were performed according to the manufacturer’s protocol using EZ:faast Amino Acid Kit (Phenomenex, Torrance, CA, USA) and a Thermo Scientific Vanquish UHPLC/TSQ Quantis triple quadrupole mass spectrometer (Thermo Electron North America, West Palm Beach, FL, USA) operated by the Emory + Children’s Pediatric Biomarkers Core (http://www.pedsresearch.org/research/cores/biomarkers-core; accessed on 4 December 2018).

#### 2.3.2. Measurement of Amino Acids, eNOS and ADMA in Myocardial Tissue

Myocardial tissue was processed using 1.5 mm bead beat homogenization on the Benchmark BeadBug Mini Homogenizer (Benchmark Scientific, Sayreville, NJ, USA). Briefly, lysis buffer was added to heart tissue for a final concentration of 50 mg/mL. The tube was homogenized at 400 beats/min for 60 s then incubated overnight at 4 degrees Celsius. The tube was subsequently sonicated at 40 Hertz for 5 min then centrifuged at 10,000× *g* for 5 min. The supernatant was then transferred for analysis of amino acid and ADMA concentrations which were all quantified according to the manufacturer’s protocol via LC-MS/MS using EZ:faast Amino Acid Kit (Phenomenex, Torrance, CA, USA) and a Thermo Scientific Vanquish UHPLC/TSQ Quantis triple quadrupole mass spectrometer (Thermo Electron North America, West Palm Beach, FL, USA) operated by the Emory + Children’s Pediatric Biomarkers Core. Myocardial eNOS concentration was measured using ELISA assay (Hycult Biotech, Wayne, PA, USA) according to the manufacturer’s protocol. Myocardial eNOS concentration was subsequently normalized to protein content.

#### 2.3.3. Measurement of Arginase Concentration and Activity

Plasma arginase concentration (ng/mL) and myocardial tissue arginase I and II concentrations (ng/mL) were measured using ELISA assay (Hycult Biotech, Wayne, PA, USA) according to the manufacturer’s protocol; myocardial tissue arginase concentrations were normalized to protein content. Arginase activity (units/L) was measured via colorimetric assay that monitors the conversion of arginine to urea and ornithine according to the manufacturer’s protocol (MilliporeSigma, St. Louis, MO, USA); myocardial tissue arginase activity was normalized to protein content.

#### 2.3.4. Statistical Analyses

Statistical analyses were conducted using SAS 9.4 (Cary, NC, USA) and statistical significance was assessed at the 0.05 level. Normality of continuous variables was assessed using histograms, normal probability plots, and the Anderson-Darling test for normality. Descriptive statistics were calculated for all variables of interest and included medians and interquartile ranges or counts and percentages, as appropriate. Chi-square tests were used to compare the distribution of categorical variables among groups. When expected cell counts were small (<5), a Fisher’s exact test was used in place of the Chi-square test. Wilcoxon rank-sum tests were used to compare continuous variables between 2 groups. When comparing continuous variables with 3 groups, Wilcoxon rank-sum tests using multiple comparisons based on pairwise rankings via the DSCF (Dwass, Steel, Critchlow-Fligner) option was used. Due to the non-normal nature of the data, a Spearman’s rank-order correlation with an associated 95% confidence interval (CI) is reported to quantify the association between various components of the arginine metabolism pathway and measures of cardiovascular structure and function. In all studies, a *p*-value of 0.05 was considered statistically significant.

## 3. Results

### 3.1. Mice with CKD Demonstrate Altered Amino Acid Metabolism Compared to Healthy Controls

Concentrations of plasma amino acids involved in arginine metabolism were compared between mice with and without CKD at 8 and 16 weeks after the onset of CKD (Table 1). Plasma citrulline concentration, the precursor to endogenous arginine synthesis, increases in mice with CKD compared to controls at both 8 and 16 weeks. Conversely, there was a non-statistically significant reduction in plasma arginine concentrations in mice with CKD compared to controls at both 8 and 16 weeks. There was also a non-statistically significant reduction in GABR in mice with CKD compared to controls at both time points. Plasma glutamine, a precursor to citrulline synthesis, is significantly reduced in mice with CKD compared to controls at 8 weeks.

### 3.2. Myocardial Dysfunction Correlates with Dysregulated Amino Acid Metabolism in Mice with and without CKD

Table 2 describes the echocardiographic findings in mice with and without CKD. Mice with CKD have increased RWT, a measure of LVH, decreased E/A ratio, a measure of diastolic dysfunction, and decreased GLS, a measure of myocardial dysfunction, at both 8 and 16 weeks compared to controls. Amino acid correlations with echocardiographic findings were conducted using Spearman correlation coefficients. In mice with and without CKD (*n* = 35), higher citrulline concentration correlated with LVH (Figure 2A), diastolic dysfunction (Figure 2B) and impaired myocardial strain (Figure 2C). While the reduction in GABR was not statistically significant, in mice with and without CKD (*n* = 35), lower GABR correlated with LVH (Figure 2D) and worsening diastolic function (Figure 2E). In CKD mice only, lower GABR correlated with higher RWT [r = −0.495; *p* = 0.031; *n* = 19) GABR also correlated with the myocardial tissue eNOS concentration in mice with and without CKD [r = 0.439; *p* = 0.013; *n* = 31]. In mice with and without CKD (*n* = 35), lower glutamine concentration correlated with LVH (Figure 2G), diastolic dysfunction (Figure 2H) and impaired myocardial strain (Figure 2I).

In a sub-group analysis of mice with and without CKD (Table 3), reduced GABR correlated inversely with higher RWT (reflective of LVH) in mice with CKD and correlated with low E/A ratio (reflective of diastolic dysfunction) in mice with CKD.

### 3.3. Plasma Arginase Activity Increases in Mice with Longer Duration of CKD

There was no difference in plasma arginase activity between mice with and without CKD at 8 weeks, however, at 16 weeks, plasma arginase activity was increased in mice with CKD compared to controls (*p* = 0.05), (Table 1), and compared to mice at 8 weeks CKD (*p* = 0.002), (Figure 3A). There was no difference in myocardial arginase I and arginase II concentration or myocardial arginase activity in mice with and without CKD at 8 or 16 weeks (Table 1). There was no correlation between plasma and myocardial arginase activity [r = 0.27, *p* = 0.113; *n* = 35], however, plasma arginase activity correlated with arginine/ornithine ratio [r = −0.523, *p* = 0.002; *n* = 33]. In mice with and without CKD (*n* = 35), higher plasma arginase activity correlated with impaired myocardial strain (Figure 3D).

### 3.4. Arginase Inhibition Improves Global Longitudinal Strain in Mice with CKD

Echocardiographic measures of LVH, diastolic dysfunction and myocardial strain were obtained in mice pre- and post-treatment with oral CB-280 treatment. Arginase inhibition resulted in improved GLS in mice with CKD (*p* = 0.03), (Figure 4C) but there was no effect on RWT or E/A ratio.

### 3.5. Amino Acid Metabolism and Arginase Activity Are Dysregulated in Children with CKD/ESRD

Banked plasma was analyzed to determine whether children with CKD have similar findings of dysregulated arginine metabolism to those observed in the mouse model of CKD. Table 4 shows the demographic and clinical characteristics of the study participants; there was no difference in age, sex, race, or ethnicity between CKD/ESRD patients and healthy controls. In CKD/ESRD patients, there was no difference in the presence of systolic or diastolic hypertension and no difference between a glomerular vs. non-glomerular etiology for CKD/ESRD (not shown).

Plasma citrulline concentration was significantly increased in children with pre-dialysis CKD (*p* = 0.019) and ESRD (*p* < 0.001) compared to healthy controls. There was a non-statistically significant reduction in GABR in children with pre-dialysis CKD and ESRD compared to controls. Plasma proline concentration was also significantly increased in children with pre-dialysis CKD (*p* = 0.021) and ESRD (*p* < 0.001) compared to healthy controls. Unlike mice with CKD, there was a non-statistically significant increase in plasma glutamine concentration in children with pre-dialysis CKD and ESRD compared to healthy controls.

Plasma arginase activity was significantly higher in children with pre-dialysis CKD compared to healthy controls (*p* = 0.047). Interestingly, plasma arginase activity was also significantly higher in children with ESRD compared to healthy controls, despite a lower concentration (*p* = 0.04). Children with ESRD were further differentiated by their modality of dialysis: hemodialysis (HD, *n* = 20) vs. peritoneal dialysis (PD, *n* = 12); Table 5. Plasma arginase concentration was lower in children on HD compared to controls (*p* = 0.04) while plasma arginase activity was higher (*p* = 0.048). Plasma arginase activity was also found to be higher in children with a glomerular etiology for CKD/ESRD vs. a non-glomerular etiology 3.8 (2.44–6.91) vs. 2.1 (1.3–3.5); *p* = 0.025. Plasma arginase activity inversely correlated with the arginine/ornithine ratio [r = −0.396, *p* = 0.0371].

We next examined the relationship between plasma arginase activity and LVH (as defined by LVMI z-score ≥2) in CKD and ESRD patients (Table 6). A total of 38 patients had echocardiograms and plasma arginase concentration, activity, and ADMA available for analysis. Of the 24 patients with LVMI z-score <2, 14 were dialysis patients (58%); all patients with LVMI z-score ≥2 were dialysis patients. Plasma arginase activity was increased in children with LVH, but this did not achieve statistical significance (*p* = 0.08). A similar trend was seen between increasing LVMI and higher plasma arginase activity [r = 0.34; *p* = 0.06; *n* = 31]. This trend persisted on adjusted analyses, when we controlled for hypertension (as defined by SBP > 95th %ile); R-squared = 0.276; *p* = 0.066; *n* = 31. The downstream metabolite of the arginase enzyme is ornithine. In children with pre-dialysis CKD and ESRD, a significant correlation was seen between increasing LVMI and higher plasma ornithine levels on Spearman correlation analysis [r = 0.34; *p* = 0.046; *n* = 33].

### 3.6. ADMA Accumulates in Mice and Children with CKD/ESRD

Plasma concentration of ADMA was elevated in both mice (Table 1) as well as children (Table 4) with CKD compared to healthy controls. In children with CKD and ESRD, higher ADMA levels correlated with left ventricular hypertrophy (increased RWT) [r = 0.54; *p* = 0.0003; *n* = 40]; this correlation persisted after controlling for HTN (SBP > 95th %ile) [R-squared = 0.15; *p* = 0.04; *n* = 40].

### 3.7. Total Nitrate/Nitrite Accumulates in Children with CKD

Plasma levels of NO_x_ (total nitrate/nitrite; mmol/L) were elevated in children with CKD and ESRD compared to healthy controls (*p* ≤ 0.001; Table 4). Interestingly, the Spearman correlation demonstrated higher plasma NO_x_ levels in patients with a lower LVMI [r = −0.52; *p* = <0.001; *n* = 38].

## 4. Discussion

Several studies have examined amino acids involved in arginine metabolism in CKD [13,14,49,50] due to arginine’s key role in nitric oxide synthesis and, therefore, its ability to contribute to cardiovascular health. Arginine is the substrate for multiple enzymatic reactions including the NOS and arginase enzymes. Arginase, which catabolizes arginine to ornithine, can reduce NO synthesis by competing with NOS for its arginine substrate. Arginase activity has been implicated in endothelial dysfunction in many disease states, including human and animal models of hypertension, pulmonary hypertension, diabetic vascular disease, sickle cell disease, thalassemia, arthritis and COVID-19 infection [22,25,35,51,52,53,54,55,56,57,58,59,60]. The NOS inhibitor, ADMA, also contributes to endothelial dysfunction by competitive inhibition of NOS and reduced NO synthesis. The most important findings in our study were the increased arginase activity seen in both children and the mouse model of CKD and the reduced GABR with correlations to myocardial dysfunction seen in the mouse model of CKD. These findings are very exciting due to the growing body of evidence showing links between reduced GABR and increased arginase activity with dysfunction of the cardiovascular system, immune system, central nervous system, and kidneys [61].

Our initial aim was to examine whether GABR, which reflects the overall status of arginine bioavailability for NO synthesis, correlated with myocardial dysfunction in CKD. Low GABR is associated with major adverse cardiovascular events and mortality in several patient populations [25,32,35,36,37,62] and is more predictive of cardiovascular risk than arginine concentration alone; this is likely because low GABR likely reflects a state of endothelial dysfunction [32]. In mice with and without CKD, GABR correlated with the tissue eNOS concentration, therefore GABR may represent a measurable assessment of endothelial function. While we only demonstrated a non-statistically significant reduction in GABR in children and the mouse model of CKD, the correlations of lower GABR with LVH and diastolic dysfunction (two highly prevalent measures of myocardial dysfunction in CKD) may be of clinical relevance in patients at risk for CVD. Specifically, low GABR could be studied as a minimally invasive marker of endothelial dysfunction in larger studies involving CKD patients.

Increasing citrulline levels in both the mouse model of CKD and children with CKD is of interest given the role of the kidney in de novo arginine synthesis from citrulline. The renal enzymes of arginine synthesis, arginosuccinate synthetase (ASS) and arginosuccinate lyase (ASL), occur in the cells of the proximal tubule and ASS/ASL abundance has been shown to be reduced in CKD animal models [31,63]. Rising citrulline levels correlate strongly with rising creatinine levels in patients with sickle cell disease (27) and may reflect the consequence of renal dysfunction on arginine synthesis with injury to the proximal tubule. Including citrulline in the assessment of global arginine bioavailability, therefore, takes into account the impact of renal dysfunction [27,28].

Plasma arginase activity is not routinely measured in all CKD studies, however, prior studies have demonstrated that arginase inhibition in a rat model of renal ablation protected the kidney from histologic damage and slowed the progression of renal failure, Ref. [64] suggesting some contribution of arginase activity to CKD progression. Interestingly, a study by Martens et al. [65] evaluating the contribution of arginase to endothelial dysfunction in the 5/6 nephrectomy rat model of CKD, found that arginase inhibition did not restore endothelium-dependent relaxation in aortic rings of rats 8 weeks after onset of CKD. These results are not surprising, however, given our findings that plasma arginase activity only significantly increased by 16 weeks of CKD in the murine experiments. This suggests that arginase’s role in dysregulated arginine metabolism occurs in more advanced stages/longer duration of CKD therefore additional studies to evaluate the temporal kinetics of plasma arginase activity during CKD are warranted. Chen et al. demonstrated unchanged arginase expression and activity in the kidney cortex, but an increase in aortic arginase expression; arginase activity was not measured due to a lack of tissue. Our study measured myocardial arginase I and II concentrations and arginase activity; interestingly, these measures did not correlate with myocardial dysfunction or plasma arginase activity. Based on these findings, it is critical to identify the source of increased plasma arginase activity (possibly vascular?) since it may inform the correlation of plasma arginase activity with myocardial dysfunction. Our study also demonstrated increased plasma arginase activity in hemodialysis patients and identified a trending association with LVH (Table 5). Increased plasma arginase activity compared to its concentration is a novel observation suggesting a mechanistic activation of arginase on hemodialysis or it is possible that an endogenous arginase inhibitor is removed during hemodialysis. Studies to identify the mechanism(s) for increased arginase activity specifically in hemodialysis are warranted.

This study found that arginase inhibition via CB-280 resulted in improved myocardial strain. GLS is an early marker of systolic dysfunction which is predictive of mortality in heart failure patients [66] and can be seen as early as 2 weeks after the onset of CKD in the partial nephrectomy mouse model [39]. Arginase inhibition was also shown to improve the progression of cardiac remodeling in uremic mice [67]. Arginine replacement therapy has been used with success in arginine deficiency states e.g., sickle cell disease, where arginine dysregulation is mediated in part by excess arginase activity released from the erythrocyte during hemolysis [33,34,68,69,70]. Arginine replacement therapy is now in phase 2 and 3 clinical trials with good safety profiles to date [70]. Early therapeutic intervention with an arginase inhibitor or with arginine supplementation may improve cardiac outcomes in CKD if its effects on myocardial strain alter the subsequent development of LVH and diastolic dysfunction.

The association of ADMA with cardiovascular complications has been demonstrated in multiple studies to date [71,72,73,74,75,76]. ADMA is considered an independent mortality and cardiovascular risk factor in CKD. Approximately 80% of ADMA is metabolized by dimethylarginine dimethylaminohydrolase (DDAH) and 20% is excreted unchanged in the urine [76]. ADMA accumulates in CKD due to decreased renal clearance and decreased catabolism by the DDAH enzyme. While ADMA is known to inhibit NOS with subsequent reduction in NO synthesis, ADMA has also been shown to cause NOS to generate oxygen-free radicals and peroxynitrite instead of NO. This results in oxidative stress which further perpetuates CV risk. Infusions of ADMA have resulted in the development of hypertension in murine studies [77] and hypertension is a known risk factor for myocardial dysfunction. Our study identified correlations between increased ADMA levels and myocardial dysfunction in children with CKD, a finding that persisted even after controlling for hypertension. The mechanisms by which ADMA mediates myocardial dysfunction are therefore likely multifactorial and further studies are warranted.

Our study has several limitations. The analyses conducted in children utilized banked plasma and echocardiograms from a repository; this resulted in a small sample size which may have affected our ability to achieve statistical significance for some measures. This also limited the echocardiographic measures that were available for analysis since post-processing of the images for new measurements (e.g., GLS) was not feasible. Moreover, plasma samples were analyzed at only a single time-point, but despite this limitation, associations with cardiovascular measures by echocardiography were still identified. For the analyses conducted in mice, we were only able to assess arginase activity and not concentration due to the limited volume of plasma available. Although arginase activity and concentration typically correlate with each other, arginase activity can be modulated by endogenous inhibitors or activators that have no impact on concentration. However, our observations in children with CKD support past studies that demonstrate a stronger association of arginase activity with clinical measures compared to its concentration [35], therefore the activity of arginase may be of more clinical relevance than its concentration. We acknowledge that diet and in vivo compartmentalization of amino acids can influence plasma amino acid levels; these are factors that we could not control for in this study, so they are study limitations. Finally, direct measurement of nitric oxide is difficult due to its rapid auto-oxidation. We measured nitric oxide metabolites utilizing the Greiss colorimetric assay, which may not be the most sensitive assay for the detection of total nitric oxide production but was the most feasible test for us within the confines of our plasma availability and funding.

## 5. Conclusions

This study highlights the complexity of arginine metabolism in CKD. Myocardial dysfunction in CKD is also quite complex and likely multifactorial. Reduced GABR and increased arginase activity are known to contribute to the development of endothelial dysfunction, and this is the likely reason for the trending correlations we identified with myocardial dysfunction. The correlations that persisted after controlling for hypertension suggest that hypertension alone is not the sole driver for myocardial dysfunction in CKD. Consequently, further studies are warranted to elucidate the potential mechanisms for increased plasma arginase activity and reduced GABR in CKD and the therapeutic impact of targeting increased arginase activity and low arginine bioavailability in CKD.

## Figures and Tables

**Figure 1 nutrients-15-02162-f001:**
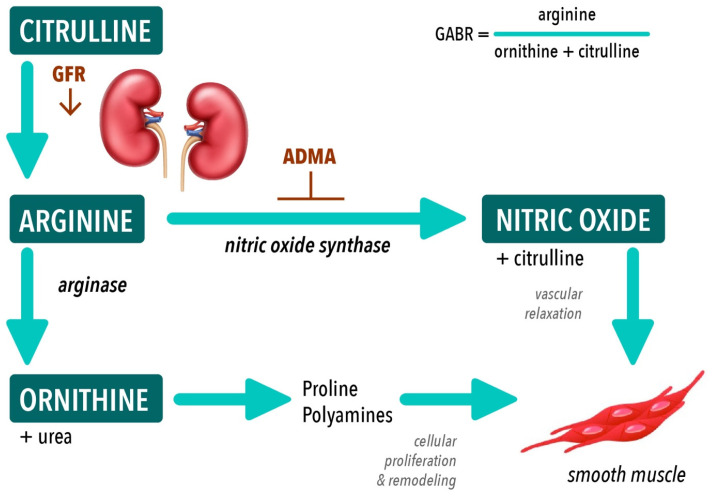
Arginine biosynthetic pathways De novo arginine synthesis occurs in the proximal tubules of the kidneys from citrulline. Arginine is the substrate for both nitric oxide synthase and arginase enzymes. Global arginine bioavailability ratio (GABR) reflects substrate availability for nitric oxide synthesis. Abbreviations: GFR, glomerular filtration rate; ADMA, asymmetric dimethylarginine.

**Figure 2 nutrients-15-02162-f002:**
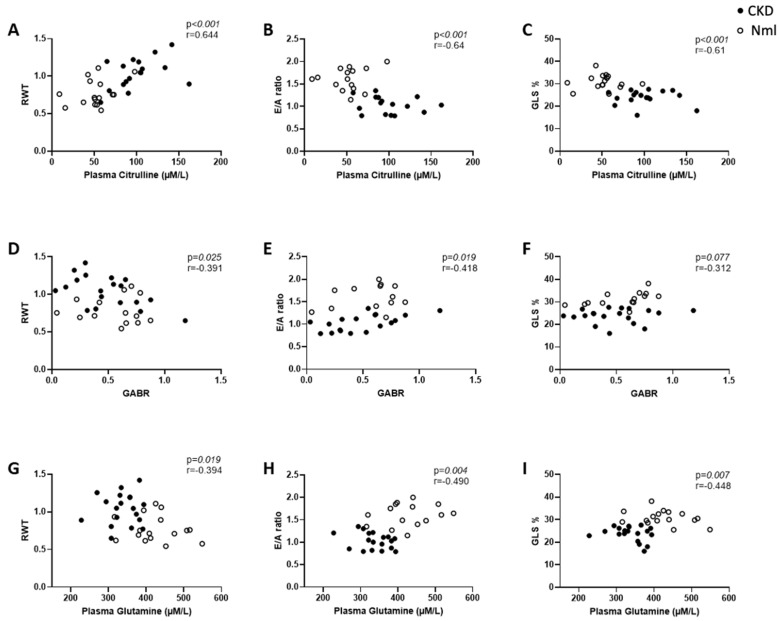
Dysregulated amino acid metabolism correlates with myocardial dysfunction in mice with and without CKD (*n* = 35). In mice with and without CKD, higher plasma citrulline concentration correlates with LVH (**A**), diastolic dysfunction (**B**), and impaired myocardial strain (**C**); reduced GABR correlates with LVH (**D**), diastolic dysfunction (**E**), but not with stain (**F**); reduced plasma glutamine concentration correlates with LVH (**G**), diastolic dysfunction (**H**), and impaired myocardial strain (**I**). Abbreviations: GABR, global arginine bioavailability ratio; RWT, relative wall thickness; CKD, chronic kidney disease; GLS, global longitudinal strain. Note: sample sizes may differ due to varying missingness in amino acid concentration and/or echo data.

**Figure 3 nutrients-15-02162-f003:**
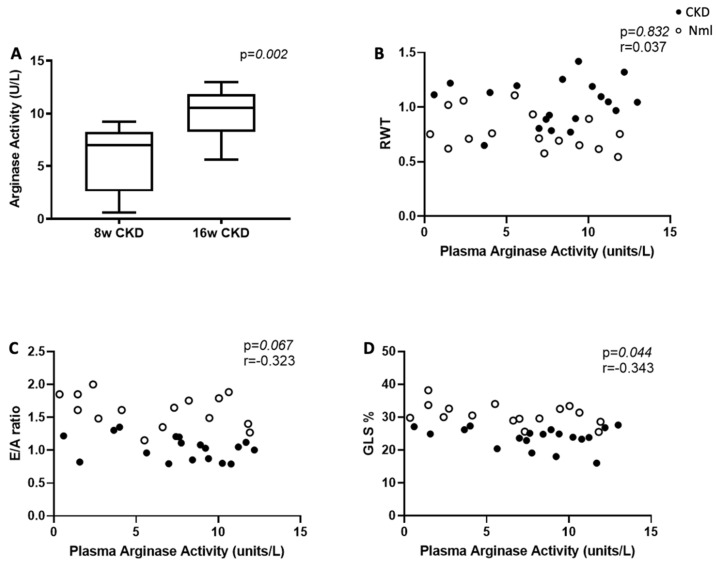
Plasma arginase activity correlates with myocardial dysfunction in mice with and without CKD (*n* = 35). In mice with CKD, plasma arginase activity is increased at 16 weeks compared to 8 weeks (**A**); no correlation between plasma arginase activity and LVH as measured by RWT in mice with and without CKD (**B**); a trending correlation is identified between plasma arginase activity and diastolic dysfunction as measured by E/A ratio in mice with and without CKD (**C**); plasma arginase activity correlates with impaired myocardial strain as measured by GLS in mice with and without CKD (**D**). Abbreviations: GABR, global arginine bioavailability ratio; RWT, relative wall thickness; CKD, chronic kidney disease; GLS, global longitudinal strain. Note: sample sizes may differ due to varying missingness in arginase activity and/or echo data.

**Figure 4 nutrients-15-02162-f004:**
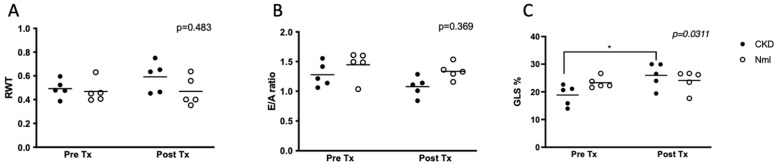
Arginase inhibition with CB-280 improves GLS in mice with CKD (*n* = 10). In mice with and without CKD, arginase inhibition via CB-280 restored impaired myocardial straI (**C**), but no effect on LVH (**A**) or diastolic function (**B**). Abbreviations: RWT, relative wall thickness; GLS, global longitudinal strain; Tx, treatment.

**Table 1 nutrients-15-02162-t001:** Amino acid metabolism in plasma and hearts of mice with (*n* = 19) and without (*n* = 16) CKD.

Measure	8 Weeks (*n* = 18)			16 Weeks (*n* = 17)		
	Control *n* = 9, 50.0%	CKD *n* = 9, 50.0%	*p*-Value	Control *n* = 7, 41.2%	CKD *n* = 10, 58.8%	*p*-Value
Plasma BUN (mg/dL)	53.7 (44.6–67.1)	43.5 (25.0–58.0)	0.224	26.7 (25.7–31.8)	54.3 (47.4–66.6)	<0.001
Plasma Cystatin C (ug/mL)	0.293 (0.232–0.376)	0.717 (0.677–0.772)	0.002	0.327 (0.292–0.414)	0.642 (0.582–0.865)	<0.001
Plasma Arginine (µM/L)	152.8 (125.8–158.9)	128.0 (89.6–163.4)	0.603	123.3 (54.3–134.7)	102.1 (70.1–117.2)	0.737
Plasma Ornithine (µM/L)	152.3 (106.7–157.6)	85.9 (75.0–117.5)	0.095	138.9 (112.3–235.5)	229.4 (144.7–274)	0.206
Plasma Citrulline (µM/L)	50.2 (37.2–53.2)	87.6 (84.3–96.0)	<0.003	57.1 (50.8–73.1)	106.0 (102.6–141.8)	0.009
GABR = Arg/(Orn+Cit)	0.75 (0.62–0.87)	0.61 (0.55–0.79)	0.664	0.64 (0.22–0.70)	0.30 (0.20–0.43)	0.176
Plasma Proline (µM/L)	160.9 (129.3–166.9)	152.3 (141.1–168.4)	0.728	135.9 (102.8–161.5)	127.0 (105.7–136.1)	0.453
Plasma Glutamate (µM/L)	97.8 (84.6–168.2)	49.0 (22.4–73.8)	0.012	84.3 (68.9–151.2)	78.5 (72.8–85.6)	0.495
Plasma Glutamine (µM/L)	411.9 (397.7–475.2)	322.3 (307.4–333.2)	<0.001	425.5 (319.5–440.1)	359.1 (334.0–374.2)	0.118
Plasma Arginase Activity (U/L)	7.3 (4.1–9.5)	7.0 (3.7–7.6)	0.438	5.5 (1.5–10.0)	10.5 (8.4–11.7)	0.052
Arg/Orn ratio	1.003 (0.641–2.343)	1.56 (1.013–1.911)	0.258	0.996 (0.269–1.168)	0.53 (0.277–0.736)	0.27
Plasma ADMA (µM/L)	7.135 (7.089–7.269)	7.188 (7.094–7.236)	0.081	7.092 (7.038–7.146)	7.171 (7.091–7.285)	0.036
Myocardial Arginase I concentration (ng/mg protein)	0.64 (0.49–0.71)	0.14 (0.09–0.67)	0.152	0.81 (0.18–1.10)	0.45 (0.24–0.79)	0.419
Myocardial Arginase II concentration (ng/mg/protein)	0.27 (0.25–0.31)	0.31 (0.17–0.33)	0.794	0.35 (0.29–0.40)	0.29 (0.16–0.41)	0.368
Myocardial Arginase Activity (units/mg protein)	16.47 (15.43–16.74)	16.74 (15.41–17.27)	0.827	14.39 (14.12–18.34)	16.74 (15.43–17.54)	0.222
Myocardial eNOS concentration (ng/mg protein)	3.22 (3.03–3.64)	4.20 (2.78–4.52)	0.728	3.47 (3.39–4.89)	4.07 (2.90–5.97)	1.000

Values reported as Median (25th–75th range) and compared using Wilcoxon rank-sum tests using multiple comparisons based on pairwise rankings. Abbreviations: BUN, blood urea nitrogen; GABR, global arginine bioavailability ratio; Arg, arginine; Orn, ornithine; Cit, citrulline; ADMA, asymmetric dimethylarginine.

**Table 2 nutrients-15-02162-t002:** Echocardiographic findings in mice with (*n* = 19) and without (*n* = 16) CKD.

Measure	8 Weeks (*n* = 18)			16 Weeks (*n* = 17)		
	Control *n* = 9, 50.0%	CKD *n* = 9, 50.0%	*p*-Value	Control *n* = 7, 41.2%	CKD *n* = 10, 58.8%	*p*-Value
RWT	0.69 (0.62–0.71)	0.90 (0.81–1.11)	0.020	0.89 (0.75–1.06)	1.14 (1.05–1.26)	0.036
E/A ratio	1.65 (1.49–1.85)	1.20 (1.03–1.22)	0.003	1.61 (1.27–1.85)	0.96 (0.85–1.05)	0.005
GLS	30.50 (29.50–32.50)	25.10 (23.60–26.20)	0.012	30.00 (28.96–33.70)	23.85 (20.40–24.90)	0.004

Values reported as Median (25th–75th range) and compared using Wilcoxon rank-sum tests using multiple comparisons based on pairwise rankings. Abbreviations: RWT, relative wall thickness; GLS, global longitudinal strain.

**Table 3 nutrients-15-02162-t003:** Subgroup Correlations for Amino Acids for Mice with (*n* = 19) and without (*n* = 16) CKD.

Measures	CKD, *n* = 19			Without CKD, *n* = 16		
	r	*n*	*p*-Value	r	*n*	*p*-Value
**Plasma Citrulline (µM/L)**						
RWT	0.419	17	0.095	0.156	16	0.564
E/A Ratio	−0.285	16	0.283	0.068	15	0.812
GLS %	−0.058	17	0.826	−0.076	16	0.780
**GABR**						
RWT	−0.495	19	0.033	−0.134	14	0.649
E/A Ratio	0.577	18	0.014	0.225	13	0.459
GLS %	0.128	19	0.601	0.763	14	0.002
**Plasma Glutamine (µM/L)**						
RWT	−0.005	19	0.986	−0.112	16	0.681
E/A Ratio	−0.329	18	0.182	0.179	15	0.524
GLS %	−0.218	19	0.371	−0.132	16	0.625
**Plasma Arginase Activity (units/L)**						
RWT	0.139	19	0.570	−0.394	16	0.131
E/A Ratio	−0.344	18	0.163	−0.311	15	0.259
GLS %	−0.129	19	0.599	−0.459	16	0.074

Note: sample sizes (*n*) differ due to varying missingness in Amino Acids, RWT, E/A Ratio, and GLS % from data.

**Table 4 nutrients-15-02162-t004:** Demographic, clinical and biochemical profile of children with and without CKD (*n* = 58).

Measure	Normal *n* = 11 (19.0%)	CKD *n* = 15 (25.8%)	ESRD *n* = 32 (55.2%)	*p*-Value Nml vs. CKD	*p*-Value Nml vs. ESRD	*p*-Value CKD vs. ESRD
** Demographics**						
Age at sample	9.9 (4.5–12.6)	11.9 (7.2–16.1)	11.5 (5.0–14.8)	0.476	0.797	0.746
N (%) male	5 (50.0%)	12 (80.0%)	22 (68.8%)	0.194	0.451	0.503
Race ^1^	-	-	-	0.515	0.791	0.056
Black	5 (50.0%)	6 (40.0%)	17 (53.1%)	-	-	-
White	4 (40.0%)	9 (60.0%)	9 (28.1%)	-	-	-
Other	1 (10.0%)	0 (0.0%)	6 (18.8%)	-	-	-
Ethnicity ^2^				*n*/a		0.399
Hispanic/Latino	*n*/a	1 (9.1%)	8 (27.6%)	-	-	-
Non-Hispanic/ Non-Latino	*n*/a	10 (90.9%)	21 (72.4%)	-	-	-
**Clinical profile ^3^**						
N (%) with SBP >95th %ile	*n*/a	5 (62.5%)	17 (65.4%)	*n*/a	*n*/a	1.000
N (%) with DBP >95th %ile	*n*/a	4 (50.0%)	14 (53.8%)	*n*/a	*n*/a	1.000
LVMI z-score	*n*/a	1.38 (0.93–1.68)	1.81 (−1.08–2.88)	*n*/a	*n*/a	0.908
**Biochemical profile**						
Arginine (µM/L)	18.3 (7.1–23.2)	14.2 (9.2–20.7)	20.9 (13.6–30.7)	0.990	0.378	0.445
Ornithine (µM/L)	30.9 (25.2–37.9)	38.5 (29.6–55.5)	38.6 (29.5–51.4)	0.191	0.349	0.989
Citrulline (µM/L)	6.5 (3.7–9.4)	15.6 (11.5–19.3)	29.1 (19.3–75.7)	0.019	<0.001	0.003
GABR = Arg/(Orn + Cit)	0.54 (0.28–0.63)	0.23 (0.15–0.35)	0.28 (0.17–0.48)	0.338	0.134	0.951
Proline (µM/L)	47.4 (38.6–63.1)	73.1 (62.5–97.1)	124.8 (78.8–188.0)	0.021	<0.001	0.116
Glutamate (µM/L)	38.8 (33.6–55.2)	42.1 (19.0–63.6)	57.0 (36.6–93.9)	0.9908	0.1937	0.2855
Glutamine (µM/L)	83.9 (69.4–110)	98.3 (77.8–120.3)	109.5 (81.8–144.5)	0.6351	0.1665	0.9016
Lysine (µM/L)	161.1 (102.1–171)	152.2 (97.7–195.2)	149.1 (−10.9–190.4)	0.9820	0.7917	0.9992
Arginase concentration (ng/mL)	18.1 (13.9–40.9)	37.7 (20.4–46.2)	11.5 (7.3–21.4)	0.749	0.072	0.016
Arginase activity (units/L)	1.52 (1.23–2.15)	3.65 (2.02–5.63)	2.92 (2.04–5.34)	0.047	0.040	0.799
Arg/Orn ratio	0.55 (0.17–0.79)	0.32 (0.23–0.49)	0.52 (0.37–0.86)	0.819	0.840	0.111
ADMA (µM/L)	7.07 (7.05–7.07)	7.5 (7.4–7.8)	8.5 (7.6–8.7)	<0.001	<0.001	0.004
NOx (µM/L)	92.9 (83.2–128.7)	174.8 (145.1–257.9)	163.9 (124.5–220.2)	<0.001	<0.001	0.603

Values reported as Median (25th–75th range) and compared using Wilcoxon rank-sum tests using multiple comparisons based on pairwise rankings. Abbreviations: %ile, percentile; SBP, systolic blood pressure; DBP, diastolic blood pressure; LVMI, left ventricular mass index; GABR, global arginine bioavailability ratio; Arg, arginine; Orn, ornithine; Cit, citrulline; ADMA, asymmetric dimethylarginine; NOx, total nitrate/nitrite. ^1^ Missing race data for 1 normal control. ^2^ Did not collect ethnicity data for normal controls; missing ethnicity data for 4 CKD and 3 ESRD patients. ^3^ Did not perform BP or echocardiograms on normal controls.

**Table 5 nutrients-15-02162-t005:** Arginase concentration and arginase activity in ESRD patients by dialysis modality (*n* = 32).

Measure	Normal (*n* = 11)	HD (*n* = 20)	PD (*n* = 12)	*p*-Value (Nml vs. HD)	*p*-Value (Nml vs. PD)	*p*-Value (HD vs. PD)
Arginase concentration (ng/mL)	18.1 (−13.9–40.9)	9.4 (6.5–21.4)	16.0 (11.2–25.3)	0.040	0.444	0.220
Arginase activity (units/L)	1.52 (1.23–2.15)	3.12 (2.10–5.34)	2.28 (1.89–4.47)	0.048	0.155	0.715

Values reported as Median (25th–75th range) and compared using Wilcoxon rank-sum tests using multiple comparisons based on pairwise rankings. Abbreviations: ADMA, asymmetric dimethylarginine; HD, hemodialysis; PD, peritoneal dialysis; Nml, normal.

**Table 6 nutrients-15-02162-t006:** Arginine metabolism and left ventricular hypertrophy in children with CKD (*n* = 38).

Measures	LVMI z-Score < 2 (*n* = 24, 63.2%)	LVMI z-Score ≥ 2 (*n* = 14, 36.8%)	*p*-Value
Arginase concentration (ng/mL)	15.8 (9.3–40.6)	8.2 (6.4–17.4)	0.084
Arginase activity (units/L)	2.57 (1.71–3.57)	5.02 (2.44–6.67)	0.083
ADMA (µM/L)	7.78 (7.36–8.51)	8.45 (7.98–8.68)	0.037

Note: sample sizes (*n*) differ from Table 4 due to varying missingness in ADMA, arginase concentration or activity from data. Values reported as Median (25th–75th range) and compared using Wilcoxon rank-sum tests using multiple comparisons based on pairwise rankings. Abbreviations: LVMI, left ventricular mass index; ADMA, asymmetric dimethylarginine.

## Data Availability

The data generated during and/or analyzed during the current study are available from the corresponding author on reasonable request.

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
