# Peer review of "Arginine Dysregulation and Myocardial Dysfunction in a Mouse Model and Children with Chronic Kidney Disease"

_nutrients, 2023, doi:10.3390/nu15092162_

Round 1
Reviewer 1 Report
In work entitled “Arginine Dysregulation and Myocardial Dysfunction in a 2 Mouse Model and Children with Chronic Kidney Disease” the authors set out to evaluate the contribution of arginine metabolism disruption to endothelial dysfunction in CKD and it concludes that arginine dysregulation correlates with myocardial dysfunction. The possible usefulness of targeting as a therapeutic tool the modulation of arginase activity is implied.
The overall work is very written, well-structured and the main aim of the study was accomplished. The methodology is appropriate and the thoroughness of the statistical analysis is of note, although some concerns have arisen regarding the correlations. The results are presented and discussed in a clear and to-the-point manner without overlooking any of the data, however some issues regarding the interpretation of the correlations in the mouse CKD model were detected. The selection of data presented in the Supplementary Materials is appropriately designed to support the main text.
Major revisions:
Main manuscript:
1) The was the authors chose to refer to the correlation analysis in the mice experiments is somewhat confusing and their interpretation of some of the results may be slightly over-optimistic, which lead to very strong statements when describing correlations between arginine metabolism and myocardial disruption.
My concern is that the correlations refer to the total of both cohorts (n = 35), encompassing both mice with and without CDK thus, the observed correlations accurately describe the relationship between parameters for the total mice population study and not specifically and independently in the Control or CKD groups. Therefore, the use of the description “in mice with (n=19) and without (n=16) CKD” is very misleading and it should be revised to refer to total mice on study.
For example, it is discussed in that “in the mouse model of CKD” statistically significant correlations were observed however, when evaluating the data presented in Figure 2, I have several reservations about if such peremptory affirmations can be made when observing the data distribution from the mice without (blank circles) and with (full circles) CKD. In graphs A, B and C there is an obvious clustering of data on the right portion of the x axis from mice without CKD and an obvious clustering of data on the left portion of the x axis from mice with CKD (and the opposite is observable for graphs G, H and I). Accordingly, when assessing correlations including all mice (both groups), statistically significant associations were found merely by statistical causality. The authors must be much more cautious when discussing these results and clearly state that the statistically significant correlations found were not specific to any of the mice groups on study (with or without CKD) and, should not infer these correlations as representing the mice CKD model, as they also include control animals.
As an alternative, I would like to propose that the authors perform a new statistical analysis of the correlations separately in each group of mice (with or without CKD). It is true that a lot of statistical power will be lost due to the decrease of n, but the tradeoff is a more accurate and specific evaluation of the associations between arginine metabolism and myocardial disruption in an unequivocally CKD study group.
2) There is a discrepancy in the number of human subjects with CKD/ESRD on study between Tables 3 and 4. In the first there is a total of 47 subjects and in the second only 38. Is this a typo? If not, it should be explained the difference in case numbers. This is particularly important concerning the correlations “in children with pre dialysis CKD and ESRD” reported in the manuscript (mainly in Results section), as a possible difference in n according to the parameters on study should be specified. For example, the correlation reported on line 348 corresponds to the same number of cases as the one reposted on line 353?
3) Throughout the manuscript text and Figures 2 and 3 – The number of cases (n) used to calculate each correlation should be added after each r and p vales. This will help with identifying which group/cohort the authors are referring to.
4) Line 437 – The authors state that “increased plasma arginase activity in hemodialysis patients and identified a trending association with LVH” and suggest that this could be related to HR treatment itself. If this is so, it is important to evaluate if the proportion of patients on HD is similar between the LVMI z-score <2 and LVMI z-score ≥2 groups and discuss this issue, as the “trending association with LVH” can possibly be the reflection of the number of individuals on HD.
Other revisions:
Main manuscript:
1) Lines 80-84 – Figure legend is cut. Please correct.
2) Line 117 – Euthanasia method should be relayed.
3) Line 118 – If serum samples were used to quantify serum urea nitrogen, how the samples were obtained should be mentioned in the previous sentence. If plasma samples were used for this measurement, this should be specified.
4) Line 147 – I believe that this sentence is misplaced and it should be deleted.
5) Line 170 – “Figure 3” should be replaced by “Figure 2”
6) Lines 191-199 – How the protein hydrolysate samples of myocardial tissue were prepared should be referenced.
7) Lines 231, 322 and 402 – Please replace “non-significant” by “non-statistically significant”, otherwise it appears that the trends observed for results are irrelevant.
8) Lines 237, 260, 268 and 289 – The n of mice without CKD must be corrected to 16.
9) Lines 240-242 – The abbreviation of BUN, Blood urea nitrogen should be added.
10) Lines 240, 263, 331 and 360 – The authors should specify “25th-75th range”.
11) Lines 267 and 287 – The symbols on the graphs should be identified (○ mice without CKD and ● mice with CKD).
12) Lines 268 and 288 – The n of mice with CKD appears as 18. Is it a typo, or correct? If correct, it should be mentioned why one is missing.
13) Line 304 – Can the authors increase the font size in the graphs of Figure 4? It is slightly difficult to perceive and does not esthetically match the other figures.
14) Lines 326 and 359 – Please replace “N” by “n” to represent the number of cases.
14) Line 326 – In Table 3, are the ADMA levels for control group correct? 7.1 (7.1 – 7.1)
15) Line 343 – Table 4 does not refer to HD or PD subjects.
16) Line 444 – Global longitudinal strain (GLS) as already been defined previously in the manuscript.
17) Line 444 – Data is not contained within the article or supplementary material, as it refers to “where raw data supporting the reported results can be found including links to publicly archived datasets analyzed or generated during the study”. The authors should amend their statement appropriately, in case such data is not publicly archived.
18) Lines 540-730 – The doi information is missing on references 12, 13, 16, 24, 28, 30, 40, 44, 45, 48 and 54.
Supplementary materials file:
1) Lines 504-505 – In the figure legend the meaning of S1 and S2 is missing.
Author Response
Major revisions:
Main manuscript:
1) The way the authors chose to refer to the correlation analysis in the mice experiments is somewhat confusing and their interpretation of some of the results may be slightly over-optimistic, which lead to very strong statements when describing correlations between arginine metabolism and myocardial disruption.
My concern is that the correlations refer to the total of both cohorts (n = 35), encompassing both mice with and without CDK thus, the observed correlations accurately describe the relationship between parameters for the total mice population study and not specifically and independently in the Control or CKD groups. Therefore, the use of the description “in mice with (n=19) and without (n=16) CKD” is very misleading and it should be revised to refer to total mice on study.
For example, it is discussed in that “in the mouse model of CKD” statistically significant correlations were observed however, when evaluating the data presented in Figure 2, I have several reservations about if such peremptory affirmations can be made when observing the data distribution from the mice without (blank circles) and with (full circles) CKD. In graphs A, B and C there is an obvious clustering of data on the right portion of the x axis from mice without CKD and an obvious clustering of data on the left portion of the x axis from mice with CKD (and the opposite is observable for graphs G, H and I). Accordingly, when assessing correlations including all mice (both groups), statistically significant associations were found merely by statistical causality. The authors must be much more cautious when discussing these results and clearly state that the statistically significant correlations found were not specific to any of the mice groups on study (with or without CKD) and, should not infer these correlations as representing the mice CKD model, as they also include control animals.
As an alternative, I would like to propose that the authors perform a new statistical analysis of the correlations separately in each group of mice (with or without CKD). It is true that a lot of statistical power will be lost due to the decrease of n, but the tradeoff is a more accurate and specific evaluation of the associations between arginine metabolism and myocardial disruption in an unequivocally CKD study group. These analyses were completed, and the manuscript updated. We chose to still include the figures including both CKD and control mice since there are no normal values for amino acid metabolites, RWT, E/A ratio or GLS in mice and the figures show clustering of CKD vs control mice on different aspects of the graphs e.g., CKD clustering to the right of the graph in Figure 2A-C where citrulline concentration is highest. While statistical significance is lost for some analyses (due to the decreased n), there is still some clinical relevance e.g., the reduced GABR that correlates with LVH and diastolic dysfunction in CKD mice.
2) There is a discrepancy in the number of human subjects with CKD/ESRD on study between Tables 3 and 4. In the first there is a total of 47 subjects and in the second only 38. Is this a typo? If not, it should be explained the difference in case numbers. This is particularly important concerning the correlations “in children with pre dialysis CKD and ESRD” reported in the manuscript (mainly in Results section), as a possible difference in n according to the parameters on study should be specified. For example, the correlation reported on line 348 corresponds to the same number of cases as the one reposted on line 353? The tables have been updated to include verbiage regarding the missingness of some data points (plasma analytes or imaging)
3) Throughout the manuscript text and Figures 2 and 3 – The number of cases (n) used to calculate each correlation should be added after each r and p vales. This will help with identifying which group/cohort the authors are referring to. The text and figures have been updated to include this.
4) Line 437 – The authors state that “increased plasma arginase activity in hemodialysis patients and identified a trending association with LVH” and suggest that this could be related to HR treatment itself. If this is so, it is important to evaluate if the proportion of patients on HD is similar between the LVMI z-score <2 and LVMI z-score ≥2 groups and discuss this issue, as the “trending association with LVH” can possibly be the reflection of the number of individuals on HD. The manuscript was edited to include this data.
Other revisions:
Main manuscript:
1) Lines 80-84 – Figure legend is cut. Please correct. We apologize; this was a formatting error and it has been rectified.
2) Line 117 – Euthanasia method should be relayed. Euthanasia was performed via CO2 asphyxiation and has been included in the manuscript.
3) Line 118 – If serum samples were used to quantify serum urea nitrogen, how the samples were obtained should be mentioned in the previous sentence. If plasma samples were used for this measurement, this should be specified. Urea nitrogen was performed on plasma; manuscript corrected to reflect this.
4) Line 147 – I believe that this sentence is misplaced and it should be deleted. Completed.
5) Line 170 – “Figure 3” should be replaced by “Figure 2” Completed
6) Lines 191-199 – How the protein hydrolysate samples of myocardial tissue were prepared should be referenced. Clarification regarding myocardial tissue processing included in manuscript.
7) Lines 231, 322 and 402 – Please replace “non-significant” by “non-statistically significant”, otherwise it appears that the trends observed for results are irrelevant. Completed.
8) Lines 237, 260, 268 and 289 – The n of mice without CKD must be corrected to 16. Completed.
9) Lines 240-242 – The abbreviation of BUN, Blood urea nitrogen should be added. Completed.
10) Lines 240, 263, 331 and 360 – The authors should specify “25th-75th range”. Completed.
11) Lines 267 and 287 – The symbols on the graphs should be identified (○ mice without CKD and ● mice with CKD). We apologize that the legend was not included in the figure; the figure has been updated.
12) Lines 268 and 288 – The n of mice with CKD appears as 18. Is it a typo, or correct? If correct, it should be mentioned why one is missing. Manuscript has been edited to reflect the accurate number of mice. In some analyses, we did not have a measurement for a specific mouse; in those situations, data is missing and we included a “missing data” statement to explain the difference in n.
13) Line 304 – Can the authors increase the font size in the graphs of Figure 4? It is slightly difficult to perceive and does not esthetically match the other figures. Figure has been edited.
14) Lines 326 and 359 – Please replace “N” by “n” to represent the number of cases. Table has been edited and “N” replaced by “n”
14) Line 326 – In Table 3, are the ADMA levels for control group correct? 7.1 (7.1 – 7.1). Table has been edited to include 2 decimal points.
15) Line 343 – Table 4 does not refer to HD or PD subjects. Manuscript has been edited appropriately.
16) Line 444 – Global longitudinal strain (GLS) as already been defined previously in the manuscript. Edited
17) Line 444 – Data is not contained within the article or supplementary material, as it refers to “where raw data supporting the reported results can be found including links to publicly archived datasets analyzed or generated during the study”. The authors should amend their statement appropriately, in case such data is not publicly archived. Completed
18) Lines 540-730 – The doi information is missing on references 12, 13, 16, 24, 28, 30, 40, 44, 45, 48 and 54. The doi has been included for the references; reference 16 does not have a doi available online
Supplementary materials file:
- Lines 504-505 – In the figure legend the meaning of S1and S2 is missing. The manuscript has been edited to include this information.
Reviewer 2 Report
1. Renal function was determined by cystatin and BUN values. I suggest to provide also the value of albuminuria in the mice groups with and without CKD.
2. The value of plasma arginina activity does not reach statistical significance (p=0.052) therefore the authors have to change the sentence at pag.8.
3. Myocardial dysfunction was measured by echocardiography, however, some cardiac biomarkers (troponins, creatin kinase or myoglobin) should be determined to support the association between myocardial dysfunction and dysregulation of amino acid metabolism in mice with and without CKD.
3.what do the black and white dots in the figure correspond to in Fig2 and Fig3 ?
4. The effect of arginase inhibitor CB-280 on echocardiographic measures is not clear and not support the results showed at fig.4. The number of mice is too small using only one dose of inhibitor. Is there any effect on cystatin values, plasma citrulline after CB.280 administration?
5. Table 3. is not a good quality and should be redone .
6. Discussion is too long and speculative.
Author Response
- Renal function was determined by cystatin and BUN values. I suggest to provide also the value of albuminuria in the mice groups with and without CKD. We appreciate the value of assessing albuminuria as another measure of renal function and will consider including this in future studies, but we are unable to perform this in the current study because we did not collect urine from mice prior to the endpoint.
- The value of plasma argininase activity does not reach statistical significance (p=0.052) therefore the authors have to change the sentence at pag.8. The manuscript has been edited to reflect this.
- Myocardial dysfunction was measured by echocardiography, however, some cardiac biomarkers (troponins, creatin kinase or myoglobin) should be determined to support the association between myocardial dysfunction and dysregulation of amino acid metabolism in mice with and without CKD. We appreciate the value of assessing plasma cardiac biomarkers as another measure of myocardial dysfunction and will consider including this in future studies, but we are unable to perform this in the current study because we do not have plasma available for additional studies.
- what do the black and white dots in the figure correspond to in Fig2 and Fig3? The figure legend was included to reflect CKD and controls.
- The effect of arginase inhibitor CB-280 on echocardiographic measures is not clear and not support the results showed at fig.4. The number of mice is too small using only one dose of inhibitor. Is there any effect on cystatin values, plasma citrulline after CB.280 administration? We attempted to obtain blood prior to treatment with the arginase inhibitor but were not able to obtain enough blood to provide enough plasma to check BUN, cystatin C, amino acids, and arginase activity. The CKD mice are less tolerant of removal of large blood volumes. Because of this limitation we were unable to evaluate pre and post treatment plasma markers adequately; we only have post treatment markers.
- Table 3. is not a good quality and should be redone. We appreciate your feedback. Are there any specific issues you would like us to address in the table?
- Discussion is too long and speculative. We appreciate your feedback. Are there any specific issues you would like us to address?
Round 2
Reviewer 1 Report
1) There is still some confusion on the text abut which cohort/group the authors are referring to when presenting correlations thus, it should be added the value of cases (n) whenever the correlations are presented on lines 259, 291, 292, 378, 393-394, 398, 407-408, 409 and 415-416. Example for line 259: [r=0.439; p=0.013; n=35].
2) The “N” should be replaced by “n” in Table 4 (line 332)
3) The sentence “In the mouse model of CKD, GABR correlated with the tissue eNOS concentration,” (lines 441-442) is incorrect since it is not in the CKD mice cohort that this correlation is observed, but rather on the mice with and without CKD cohort (result presented on line 259). The authors should change to beginning of the sentence to accurately represent the presented results.
4) The Data Availability Statement is still incorrect (line 571) because the authors only show the mean ± SD or mean (IQR) values of the data on the manuscript. As this statement refers to raw data, which is not presented within the article or supplementary material, nor is it publicly archived and available for consultation, the statement must be amended.
Author Response
1) There is still some confusion on the text abut which cohort/group the authors are referring to when presenting correlations thus, it should be added the value of cases (n) whenever the correlations are presented on lines 259, 291, 292, 378, 393-394, 398, 407-408, 409 and 415-416. Example for line 259: [r=0.439; p=0.013; n=35]. The sample sizes have been included for all correlations. Due to some missingness in the availability of echo data and plasma analytes, the sample sizes do vary; e.g. some mice didn't have enough plasma for arginase activity. Thank you for encouraging me to include these.
2) The “N” should be replaced by “n” in Table 4 (line 332) Completed.
3) The sentence “In the mouse model of CKD, GABR correlated with the tissue eNOS concentration,” (lines 441-442) is incorrect since it is not in the CKD mice cohort that this correlation is observed, but rather on the mice with and without CKD cohort (result presented on line 259). The authors should change to beginning of the sentence to accurately represent the presented results. This statement has been revised.
4) The Data Availability Statement is still incorrect (line 571) because the authors only show the mean ± SD or mean (IQR) values of the data on the manuscript. As this statement refers to raw data, which is not presented within the article or supplementary material, nor is it publicly archived and available for consultation, the statement must be amended. This statement has been amended.
Reviewer 2 Report
The paper is improved and the writing is more clear. The authors respond to all the questions.
Author Response
Thank you.